# RECALL TRACES: BACKTRACKING MODELS FOR EFFICIENT REINFORCEMENT LEARNING

**Anirudh Goyal[1], Philemon Brakel[2], William Fedus[1], Soumye Singhal[1,4],**
**Timothy Lillicrap[2], Sergey Levine[5], Hugo Larochelle[3], Yoshua Bengio[1]**

## ABSTRACT

In many environments only a tiny subset of all states yield high reward. In these cases, few of the interactions with the environment provide a relevant learning signal. Hence, we may want to preferentially train on those high-reward states and the probable trajectories leading to them. To this end, we advocate for the use of a *backtracking model* that predicts the preceding states that terminate at a given high-reward state. We can train a model which, starting from a high value state (or one that is estimated to have high value), predicts and samples which (state, action)-tuples may have led to that high value state. These traces of (state, action) pairs, which we refer to as Recall Traces, sampled from this backtracking model starting from a high value state, are informative as they terminate in good states, and hence we can use these traces to improve a policy. We provide a variational interpretation for this idea and a practical algorithm in which the backtracking model samples from an approximate posterior distribution over trajectories which lead to large rewards. Our method improves the sample efficiency of both on- and off-policy RL algorithms across several environments and tasks.

## 1 INTRODUCTION

Training control algorithms efficiently from interactions with the environment is a central issue in reinforcement learning (RL). Model-free RL methods, combined with deep neural networks, have achieved impressive results across a wide range of domains (Lillicrap et al., 2015; Mnih et al., 2016; Silver et al., 2016). However, existing model-free solutions lack *sample efficiency*, meaning that they require extensive interaction with the environment to achieve these levels of performance.

Model-based methods in RL can mitigate this issue. These approaches learn an unsupervised model of the underlying dynamics of the environment, which does not necessarily require rewards, as the model observes and predicts state-to-state transitions. With a well-trained model, the algorithm can then simulate the environment and look ahead to future events to establish better value estimates, without requiring expensive interactions with the environment. Model-based methods can thus be more sample efficient than their model-free counterparts, but often do not achieve the same asymptotic performance (Deisenroth & Rasmussen, 2011a; Nagabandi et al., 2017).

In this work, we propose a method that takes advantage of unsupervised observations of state-to-state transitions for increasing the sample efficiency of current model-free RL algorithms, as measured by the number of interactions with the environment required to learn a successful policy. Our idea stems from a simple observation: given a world model, finding a path between a starting state and a goal state can be done either forward from the start or backward from the goal. Here, we explore an idea for leveraging the latter approach and combining it with model-free algorithms. This idea is particularly useful when rewards are sparse. High-value states are rare and trajectories leading to them are particularly useful for a learner.

The availability of an exact backward dynamics model of the environment is a strong and often unrealistic requirement for most domains. Therefore, we propose learning a backward dynamics model, which we refer to as a *backtracking model*, from the experiences performed by the agent. This backtracking model $p(s_t, a_t | s_{t+1})$, is trained to predict, given a state $s_{t+1}$, which state $s_t$ the

[1]Mila, University of Montreal, [2] Google Deepmind, [3] Google Brain, [4] IIT Kanpur, [5] University of California, Berkeley. Corresponding author :anirudhgoyal19119@gmail.com

agent visited before $s_{t+1}$ and what action $a_t \sim \pi$ was performed in $s_t$ to reach $s_{t+1}$. Specifically, this is a model which, starting from a future high-value state, can be used to recall traces that have ended at this high value state, that is sequences of (state, action)-tuples. This allows the agent to simulate and be exposed to alternative possible paths to reach a high value state. A final state may be a previously experienced high-value state or a goal state may be explicitly given, or even produced by the agent using a generative model of high-value states (Held et al., 2017).

Our hypothesis is that using a backtracking model in this way should benefit learning, especially in the context of weak or sparse rewards. Indeed, in environments or tasks where the agent receives rewards infrequently, it must leverage this information effectively and efficiently. Exploration methods have been employed successfully (Bellemare et al., 2016; Held et al., 2017; Ostrovski et al., 2017) to increase the frequency at which novel states are discovered. Our proposal can be viewed as a special kind of simulated exploration proceeding backward from presumed high-value states, in order to discover trajectories that may lead to high rewards. A backtracking model aims to augment the experience of the trajectory $\tau$ leading to a high-value state by generating other

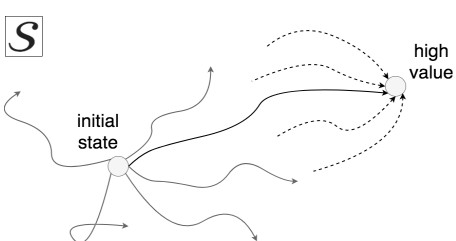

Figure 1: The policy explores the state space $\mathcal{S}$ from an initial state. Discovered high value states are then passed to the backtracking model (dashed-lines) to generate new traces that may have led to this high value state.

possible traces $\tilde{\tau}$ that could have also caused it. To summarize: the main contribution of this paper is an RL method based on the use of a backtracking model, which can easily be integrated with existing on- and off-policy techniques for reducing sample complexity. Empirically, we show with experiments on eight RL environments that the proposed approach is more sample efficient.

## 2 PRELIMINARIES

We consider a Markov decision process (MDP) defined by the tuple $(\mathcal{S}, \mathcal{A}, P, r, \gamma)$, where the state space $\mathcal{S}$ and the action space $\mathcal{A}$ may be discrete or continuous. The learner is not explicitly given the environment transition probability $p(s_{t+1}|s_t, a_t)$ for going from $s_t \in \mathcal{S}$ to $s_{t+1} \in \mathcal{S}$ given $a_t \in \mathcal{A}$, but samples from this distribution are observed. The environment emits a bounded reward $r : \mathcal{S} \times \mathcal{A} \to [r_{min}, r_{max}]$ on each transition and $\gamma \in (0, 1)$ is the discount factor. Let $\pi$ denote a stochastic policy over actions given states, and let $R(\pi) = \mathbb{E}_\pi \left[ \sum_{t=0}^T \gamma^t r(s_t) \right]$ denote the expected total return when policy $\pi$ is followed. The standard objective in reinforcement learning is to maximize the discounted total return $R(\pi)$. Throughout the text we will refer to *experienced* trajectories as $\tau = (s_1, a_1, \cdots, s_T, a_T)$ and we will refer to *simulated* experiences as traces $\tilde{\tau}$.

### 2.1 BACKTRACKING MODEL

We introduce the backtracking model $B_\phi = q_\phi(s_t, a_t|s_{t+1})$, which is a density estimator of the joint probability distribution over the previous $(s_t, a_t)$-tuple parameterized by $\phi$. This distribution is produced by both a learned backward policy $\pi_b = q(a_t|s_{t+1})$ and a state generator $q(s_t|a_t, s_{t+1})$. The backward policy predicts the previous action $a_t$ given the resulting state $s_{t+1}$. The state generator estimates the probability of a previous state $s_t$ given the tuple $(a_t, s_{t+1})$. With these models, we may decompose $q_\phi(s_t, a_t|s_{t+1})$ as $q(s_t|a_t, s_{t+1})q(a_t|s_{t+1})$.

However, for training stability with continuous-valued states, we model the density of state variation $\Delta s_t = s_t - s_{t+1}$ rather than the raw $s_t$. Therefore, our density models are given by

$$q_\phi(\Delta_t, a_t|s_{t+1}) = q(\Delta s_t|a_t, s_{t+1})q(a_t|s_{t+1}). \tag{1}$$

Note that for readability, we will drop the $\phi$-subscript unless it is necessary for clarity.

**Generating Recall Traces.** Analogous to the use of forward models in (Oh et al., 2015; Chiappa et al., 2017; Weber et al., 2017), we may generate a recall trace auto-regressively. To do so, we begin with a state $s_{t+1}$ and sample $a_t \sim q(a_t|s_{t+1})$. The state generator can then be sampled to produce

the change in state $\Delta s_t \sim q(\Delta s_t|a_t, s_{t+1})$. We can continue to unroll this process, repeating with state $s_t = \Delta s_t + s_{t+1}$ for a desired number of steps. These generated transitions are then stored as a potential trace $\tilde{\tau}$ which terminates at some final state. The backtracking model $B_\phi$ is learned by maximum likelihood, using the policy's trajectories as observations, as described in Section 3.1.

**Producing Intended High Value States.** Before recursively sampling from the backtracking model, we need to obtain presumed high-value states. Generally, such states will not be known in advance. However, as the agent learns, it will visit states $s_t$ with increasingly high value $V^\pi(s_t = s) = E_\pi[\sum_t \gamma^t r(s_t)| s_t = s]$. The agent's full experience is maintained in a replay buffer $\mathcal{B}$, in the form of tuples of $(s_t, a_t, s_{t+1}, r_t)$. Filtering of trajectories based on the returns is done, so that only top $k_{traj}$ are added to the buffer. In this work, we will investigate our approach with two methods for generating the initial high-value states.

The first method relies on picking the most valuable states stored in the replay buffer $\mathcal{B}$. As before, a valuable state may be defined by its estimated expected return $V^\pi(s)$ as computed by a critic (our off-policy method) or state that received a high reward (our on-policy method).

The second method is based on Goal GAN, recently introduced by (Held et al., 2017) where goal states $g$ are produced via a Generative Adversarial Network (Goodfellow et al., 2014). In our variant, we map the goal state $g$ to a valid point in state space $s$ using a 'decoder' $D$. For the point-mass, the goal and state are identical; for Ant, we use a valid random joint-angle configuration at that goal position. The backtracking model is then used to find plausible trajectories that terminate at that state. For both methods, as the learner improves, one would expect that on average, higher value states are used to seed the recall traces.

## 3 IMPROVING POLICIES WITH BACKTRACKING MODEL

In this section, we describe how to train the backtracking model and how it can be used to improve the efficiency of the agent's policy and aid with exploration.

### 3.1 TRAINING THE BACKTRACKING MODEL

We use a maximum likelihood training loss for training of the backtracking model $B_\phi$ on the top $k\%$ of the agent's trajectories stored in the state buffer $\mathcal{B}$. At each iteration, we perform stochastic gradient updates based on agent trajectories $\tau$, with respect to the following objective:

$$\mathcal{L}_\mathcal{B} = \log q_\phi(\tau) = \log \prod_{t=0}^{T} q(\Delta s_t, a_t|s_{t+1}) \quad = \sum_{t=0}^{T} \log q(a_t|s_{t+1}) + \log q(\Delta s_t|a_t, s_{t+1}), \quad (2)$$

where $s_t = \Delta s_t + s_{t+1}$ and $T$ is the episode length. For our chosen backtracking model, this implies a mean-squared error loss (i.e. corresponding to a conditional Gaussian for $\Delta s_t$) for continuous action tasks and a cross-entropy loss (i.e. corresponding to a conditional Multinoulli for $s_t$ given $a_t$ and $s_{t+1}$) for discrete action tasks. The buffer is constantly updated with recent experiences and the backtracking model is trained online in order to encourage generalization as the distribution of trajectories in the buffer evolves.

### 3.2 IMPROVING THE POLICY FROM THE RECALL TRACES

We now describe how we use the recall traces $\tilde{\tau}$ to improve the agent's policy $\pi_\theta$. In brief, the traces $\tilde{\tau}$ generated by the backtracking model are used as observations for imitation learning (Pomerleau, 1989; ros, 2011; Bojarski et al., 2016) by the agent. The backtracking model will be continuously updated as new actual experiences are generated, as described in Section 3.1. Imitation learning of the policy is performed simply by maximizing the log-probability of the agent's action $a_t$ given $s_t$ given by

$$\mathcal{L}_\mathcal{I} = \sum_{t=0}^{T} \log p(a_t|s_t) = \sum_{t=0}^{T} \log \pi_\theta(a_t|s_t), \quad (3)$$

where $(s_t, a_t)$-tuples come from a generated trace $\tilde{\tau}$.

Our motivation for having the agent imitate trajectories from the backtracking model is two-fold:

---

**Algorithm 1** Improve Policy via Recall Traces and Backtracking Model

---

**Require:** RL algorithm with parameterized policy (i.e. TRPO, Actor-Critic)
**Require:** Agent policy $\pi_\theta(a|s)$
**Require:** Backtracking model $B_\phi = q_\phi(\Delta s_t, a_t|s_{t+1})$
**Require:** Critic $V(s)$
**Require:** $k$ quantile of best state values used to train backtracking model, $k_{traj}$ number of trajectories filtered by returns.
**Require:** $N$; number of backward trajectories per target state
**Require:** $\alpha, \beta$; forward, backward learning rates
  1: Randomly initialize agent policy parameters $\theta$
  2: Randomly initialize backtracking model parameters $\phi$
  3: **for** $t = 1$ to $K$ **do**
  4:     Execute policy to produce trajectory $\tau$
  5:     Add trajectory $\tau = (s_1, a_1, r_1, \cdots, s_T, a_T, r_T)$ in $\mathcal{B}$
  6:     Estimate $\nabla_\theta R(\pi_\theta)$ from RL algorithm
  7:     $\theta \leftarrow \theta + \alpha \nabla_\theta R(\pi_\theta)$
  8:     Compute $\mathcal{L}_\mathcal{B}$ via Equation 2, using top $k\%$ valuable states from top $k_{traj}$ trajectories in $\mathcal{B}$
  9:     $\phi \leftarrow \phi + \beta \nabla_\phi \mathcal{L}_\mathcal{B}$
 10:     Obtain target high value state $s$ (see Algorithm 2 for details)
 11:     Generate $N$ recall traces $\tilde{\tau}$ for $s$ using $B_\phi(s)$
 12:     Compute imitation loss $\mathcal{L}_\mathcal{I}$ via Equation 3
 13:     $\theta \leftarrow \theta + \alpha \nabla_\theta \mathcal{L}_\mathcal{I}$
 14: **end for**

---

**Dealing with sparse rewards** States with significant return are emphasized by the backtracking model, since the traces it generates are initialized at high value states. We expect this behaviour to help in the context of sparse or weak rewards.

**Aiding in exploration** The backtracking model can also generate new ways to reach high-value states. So even if it cannot directly discover new high-value states, it can at least point to new ways to reach known high value states, thus aiding with exploration.

## 4 Variational Interpretation

Thus far, we have motivated the use of a backtracking model intuitively. In this section, we provide a motivation relying on a variational perspective of RL and ideas from the wake-sleep algorithm Hinton et al. (1995).

Let $R$ be the return of a policy trajectory $\tau$, i.e. the sum of discounted rewards under this trajectory. Consider the event of the return $R$ being larger than some threshold $L$. The probability of that event under the agent's policy is $p(R > L) = \sum_\tau p(R > L|\tau)p(\tau)$, where $p(\tau)$ is distribution of trajectories under policy $\pi$, $p(R > L|\tau) = 1_{R>L}$ and $1_A$ is the indicator function that is equal to 1 if $A$ is true and is otherwise 0.

Let $q(\tau)$ be any other distribution over trajectories, then we have the following classic relationship between the marginal log-probability of an observation ($R > L$) and the KL-divergence between $q$ and the posterior over a latent variable ($\tau$):

$$\log p(R > L) = \mathcal{L} + \mathrm{KL}(q(\tau)||p(\tau|R > L)) \geq \mathcal{L} \qquad (4)$$

where

$$\mathcal{L} = \sum_\tau q(\tau) \log\left(p(R > L|\tau)p(\tau)/q(\tau)\right). \qquad (5)$$

This suggests an EM-style training procedure, that alternates between training the variational distribution $q(\tau)$ towards the posterior $p(\tau|R > L)$ and training the policy to maximize $\mathcal{L}$. In this context, we view the backtracking model and the high-value states sampler as providing $q(\tau)$ implicitly. Specifically, we assume that $q$ factorizes temporally as in Equation 2 with the backtracking model providing $q(a_t|s_{t+1})$ and $q(\Delta s_t|a_t, s_{s+1})$. We parameterize the approximate posterior in this way so that we can take advantage of a model of the backwards transitions to conveniently sample

from $q$ starting from a high-value final state $s_T$. This makes sense in the context of sparse rewards, where few states have significant reward. If we have a way to identify these high-reward states, then it is much easier to obtain these posterior trajectories by starting from them.

Training $q(\tau)$ by minimizing the $\text{KL}(q(\tau)||p(\tau|R > L))$ term is hard due to the direction of the KL-divergence. Taking inspiration from the wake-sleep algorithm (Hinton et al., 1995), we instead minimize the KL in the opposite direction, $\text{KL}(p(\tau|R > L)||q(\tau))$. This can be done by sampling trajectories from $p(\tau|R > L)$ (e.g. by rejection sampling, keeping only the forward-generated trajectories which lead to $R > L$) and maximizing their log-probability under $q(\tau)$. This recovers our algorithm, which trains the backtracking model on high-return trajectories generated by the agent.

So far we have assumed a known threshold $L$. However, in practice the choice of $L$ is important. While ultimately we would want $L$ to be close to the highest possible return, at the early stages of training it cannot be, as trajectories from the agent are unlikely to reach that threshold. A better strategy is to gradually increase $L$. One natural way of doing this is to use the *top few percentile trajectories* sampled by the agent for training $q(\tau)$, instead of explicitly setting $L$. This approach can be thought of as providing a curriculum for training the agent that is adapted to its performance. This is also related to evolutionary methods (Hansen, 2016; Baluja, 1994), which keep the "fittest" samples from a population in order to re-estimate a model, from which new samples are generated.

This variational point of view also tells us how the prior over the last state should be constructed. The ideal prior $q(s_T)$ is simply a generative model of the final states leading to $R > L$. Both methods proposed to estimate $q(s_T)$ with this purpose, either non-parametrically (with the forward samples for which $R > L$) or parametrically (with generative model trained from those samples). Also, if goal states are known ahead of time, then we can set $L$ as the reward of those states (minus a small quantity) and we can seed the backwards trajectories from these goal states. In that case the variational objective used to train the policy is a proxy for log-likelihood of reaching a goal state.

## 5 RELATED WORK

**Control as inference** The idea of treating control problems as inference has been around for many years (Stengel, 1986; Kappen et al., 2012; Todorov, 2007; Toussaint, 2009; Rawlik et al., 2012). A good example of this idea is to use Expectation Maximization (EM) for RL (Dayan & Hinton, 1997), of which the PoWER algorithm (Kober et al., 2013) is one well-known practical implementation. In EM algorithms for RL, learning is divided between the estimation of the expectation over the trajectories conditioned on the reward observations and estimation of a new policy based on these expectation estimates. While we don't explicitly try to estimate these expectations, one could argue that the samples from the backtracking model serve a similar purpose. Variational inference has also been proposed for policy search (Neumann et al., 2011; Levine & Koltun, 2013). Probabilistic views of the RL problem have also been used to construct maximum entropy methods for both regular and inverse RL (Haarnoja et al., 2017; Ziebart et al., 2008).

**Using off-policy trajectories** By incorporating the trajectories of a separate backtracking model, our method is similar in spirit to approaches which combine on-policy learning algorithms with off-policy samples. Recent examples of this, like the interpolated policy gradient (Gu et al., 2017), PGQ (O'Donoghue et al., 2016) and ACER (Wang et al., 2016), combine policy gradient learning with ideas for off-policy learning and methodology inspired by Q-learning. Our method differs by using the backtracking model to obtain off-policy trajectories and is, as an idea, independent of the specific model-free RL method it is combined with. Our work to effectively propagate value updates backwards is also related to the seminal work of prioritized sweeping (Moore & Atkeson, 1993).

**Model-based methods** A wide range of model-based RL and control methods have been proposed in the literature (Deisenroth et al., 2013). PILCO (Deisenroth & Rasmussen, 2011b), is a model-based policy search method to learn a probabilistic model of dynamics and incorporate model uncertainty into long-term planning. The classic Dyna (Sutton) algorithm was proposed to take advantage of a model to generate simulated experiences, which could be included in the training data for a model-free algorithm. This method was extended to work with deep neural network policies, but performed best with models that were not neural networks (Gu et al., 2016b). Other extensions to Dyna have also been proposed (Silver et al., 2008; Kalweit & Boedecker; Heess et al., 2015).

Other approaches have also been proposed to combine advantages of both value and policy-based approaches (Nachum et al., 2017; Sukhbaatar et al., 2017). Finally, (Edwards et al., 2018) is concurrent

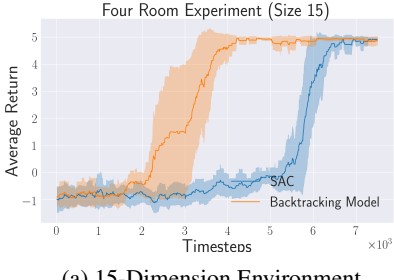
(a) 15-Dimension Environment

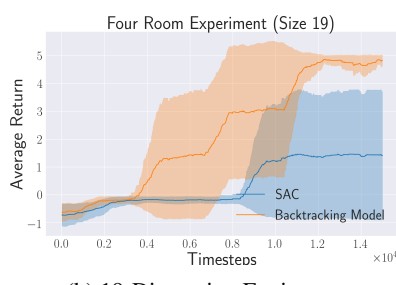
(b) 19-Dimension Environment

Figure 2: Training curves from the Four Room Environment for the Actor-Critic baseline (blue) and the backtracking model augmented Actor-Critic (orange). For the size-19 environment, several of the Actor-Critic baselines failed to converge, whereas the augmented recall trace model always succeeded in the number of training steps considered. For additional results see Figure 9 in Appendix.

work to this and also proposes training on imagined reversal steps from known goal states. (Goyal et al., 2017) proposed to use similar learning rule in the context of generative models to modify the parameters of transition operator to make the reverse of this heated trajectory more likely under a reverse cooling process.

# 6 EXPERIMENTAL RESULTS

Our experimental evaluation aims to understand whether our method can improve the sample complexity of off-policy as well as on-policy RL algorithms. Practically, we must choose the length of generated backward traces. Longer traces become increasingly likely to deviate significantly from the traces that the agent can generate from its initial state. Therefore, in our experiments, we sample fairly short traces $\tilde{\tau}$ from the backtracking model, where the length is adjusted manually based on the time-scale of each task.

**We empirically show the following results across different experimental settings:**

- Samples from the true backtracking model can be used to improve sample efficiency.
- Using a learned backtracking model starting from high value states accelerates learning for off-policy as well as on-policy experiments.
- Modeling parametrically and generating high value states (using GoalGAN) also helps.

## 6.1 ACCESS TO TRUE BACKTRACKING MODEL

Here, we aim to check if in the ideal case when the true backtracking model is known, the proposed approach works. To investigate this, we use the four-room environment from Schaul et al. (2015) of various dimensions. The 4-room grid world is a simple environment where the agent must navigate to a goal position to receive a positive reward through bottleneck states (doorways). We compare the proposed method to the scenario where the policy is trained through the actor-critic method Konda (2002) with generalized advantage estimation (GAE) Schulman et al. (2015b)

Finding the goal state becomes more challenging as the dimension increases due to sparsity of rewards. Therefore, we expect that the backtracking model would be a more effective tool in larger environments. Confirming this hypothesis, we see in Figure 2 that, as we increase the dimensionality of the maze, sample efficiency increases thanks to recall traces, compared to our baseline.

## 6.2 COMPARISON WITH PRIORITIZED EXPERIENCE REPLAY

Here we aim to compare the performance of recall traces with Prioritized Experience Replay (PER). PER stores the past experiences in a buffer and then selectively trains on high value experiences. We again use the Four-room Environment. PER gives an optimistic bias to the critic, and while it allows the reinforcement of sparse rewards, it also converges too quickly to an exploitation mode, which

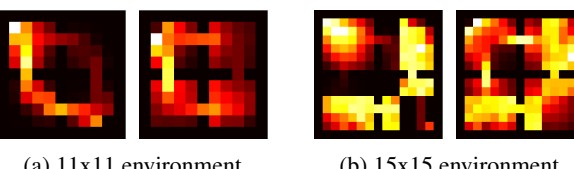

      (a) 11x11 environment        (b) 15x15 environment

Figure 3: Visitation count visualization of trained policies for PER (left) and Recall Traces (right) for two 4-room grid sizes.

can be difficult to get out of. In order to show this, we plot the state visitation counts of a policies trained with PER and recall traces and see that the latter visit more states.

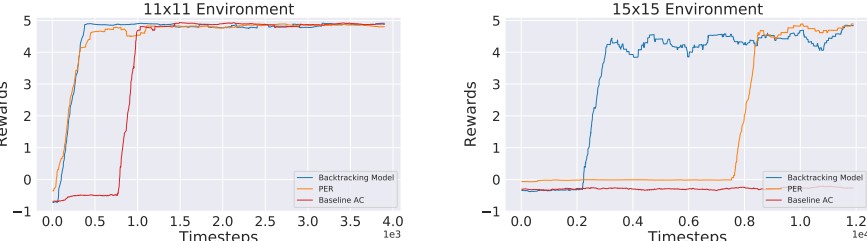

Figure 4: Plots for reward vs. time steps, comparing the performance of recall traces (labeled Back-trackingModel), PER and baseline Actor Critic (AC).

We show in Figure 4 that while PER is competitive in the smaller 11x11 environment, recall traces outperform it in the larger 15x15 environment. Figure 3 also shows how the use of a backtracking model and recall traces pushes the policy to visit a wider variety of grid positions than PER.

## 6.3 LEARNED BACKTRACKING MODEL FROM GENERATED STATES

One situation in which states to start the backtracking model at are naturally available, is when the method is combined with an algorithm for sub-goal selection. We chose to investigate how well the backtracking model can be used in conjunction with the automatic goal generation algorithm from Held et al. (2017), Goal GAN. It uses a Generative Adversarial Network to produce sub-goals at the appropriate level of difficulty for the agent to reach. As the agent learns, new sub-goals of increasing difficulty are generated. This way, the agent is pressured to explore and learn to be able to reach any location in the state space. We hypothesize that the backtracking model should help the agent to reach the sub-goals faster and explore more efficiently.

Hence, in this learning scenario, what changes is that high value states are now generated by Goal GAN instead of being selected by a critic from a replay buffer.

We performed experiments on the U-Maze Ant task described in Held et al. (2017). It is a challenging robotic locomotion task where a quadruped robot has to navigate its center of mass within some particular distance of a target goal. The objective is to cover as much of the space of the U-shaped maze as possible. We find that using the backtracking model improves data efficiency, by reaching a coverage of more than 63% in 155 steps, instead of 275 steps without it (Fig 5). More visualizations and learning curves for U-Maze Ant task as well as the N-Dimensional Point Mass task can be found in Appendix (Figs. 11, 12 and 13).

## 6.4 LEARNED BACKTRACKING MODEL - ON-POLICY CASE

We conducted robotic locomotion experiments using the MuJoCo simulator (Todorov et al., 2012). We use the same setup as (Nagabandi et al., 2017). We compare our approach with a pure model-free method on standard benchmark locomotion tasks, to learn the fastest forward-moving gait possible. The model-free approach we consider is the `rllab` implementation of trust region policy optimization (TRPO) Schulman et al. (2015a). For the TRPO baseline we use the same setup as Nagabandi et al. (2017). See the appendix for the model implementation details.

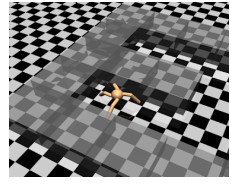 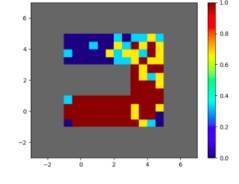 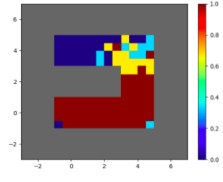

(a) U-Maze Ant Environment          (b) 275 steps: 63% coverage          (c) 155 steps: 64% coverage

Figure 5: Visualization of GoalGAN baseline (b) vs backtracking model (c) policy performance for different parts of the state space for Ant Maze task. Red indicates complete success; blue indicates failure. Backtracking model achieves equal coverage rates in fewer steps of training.

**Performance across tasks:** The results in Figure 6 show that our method consistently outperforms TRPO on all of the benchmark tasks in terms of final performance, and learns substantially faster.

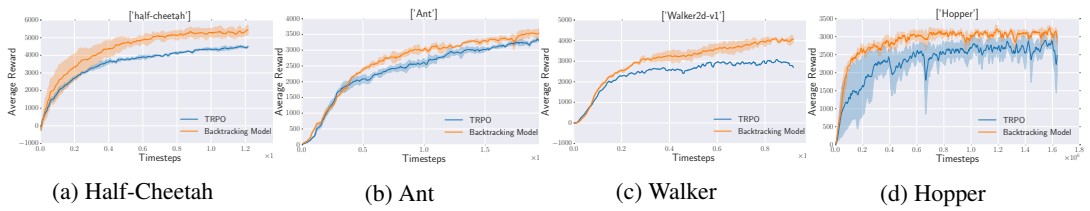

(a) Half-Cheetah          (b) Ant          (c) Walker          (d) Hopper

Figure 6: Our model as compared to TRPO. For TRPO baselines, except walker, we ran with 5 different random seeds. For our model, we ran with 5 different random seeds.

## 6.5 LEARNED BACKTRACKING MODEL - OFF POLICY CASE

Here, we evaluate on the same range of challenging continuous control tasks from the OpenAI gym benchmark suite. We compare to Soft Actor Critic (SAC) (Haarnoja et al., 2018), shown to be more sample efficient compared to other off-policy algorithms such as DDPG (Lillicrap et al., 2015) and which consistently outperforms DDPG. Part of the reason for choosing SAC instead of DDPG was that the latter is also known to be more sensitive to hyper-parameter settings, limiting its effectiveness on complex tasks Henderson et al. (2017). For SAC, we use the same hyper-parameters reported in Haarnoja et al. (2018). Implementation details for our model are listed in the Appendix.

**Performance across tasks -** The results in Figure 7 show that our method consistently improves the performance of SAC on all of the benchmark tasks, leading to faster learning. In fact, the largest improvement is observed on the hardest task, Ant.

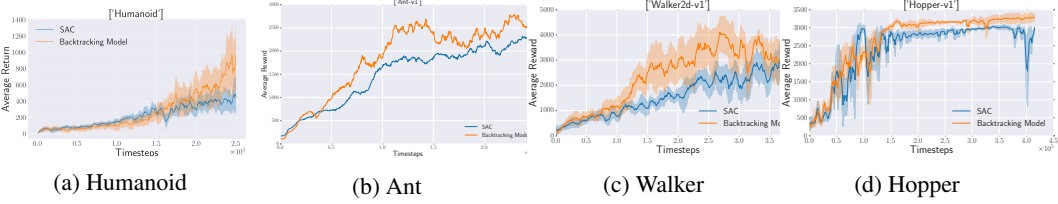

(a) Humanoid          (b) Ant          (c) Walker          (d) Hopper

Figure 7: Our model as compared to SAC. We ran SAC baselines with 2 different random seeds. For our model, we ran with 5 different random seeds.

## 7 DISCUSSION

We advocate for the use of a backtracking model for improving sample efficiency and exploration in RL. The method can easily be combined with popular RL algorithms like TRPO and soft actor-critic. Our results indicate that the recall traces generated by such models are able to accelerate learning on a variety of tasks. We also show that the method can be combined with automatic goal

generation. The Appendix provides more analysis on the sensitivity of our method to various factors and ablations. We show that a random model is outperformed by a trained backtracking model, confirming its usefulness and present plots showing the effect of varying the length of recall traces.

For future work, while we observed empirically that the method has practical value and could relate its workings from a variational perspective, but more could be done to improve our theoretical understanding of its convergence behavior and what kind of assumptions need to hold about the environment. It would also be interesting to investigate how the backtracking model can be combined with forward models from a more conventional model-based system.

## 8 ACKNOWLEDGEMENTS

The authors acknowledge the important role played by their colleagues at Mila throughout the duration of this work. The authors would like to thank Alessandro Sordoni for being involved in the earlier phase of the project, and for the Four Room Code. AG would like to thank Doina Precup, Matthew Botvinick, Konrad Kording for useful discussions. William Fedus would like to thank Evan Racah and Valentin Thomas for useful discussions and edits. The authors would also like to thank Nicolas Le Roux, Rahul Sukthankar, Gatan Marceau Caron, Maxime Chevalier-Boisvert, Justin Fu, Nasim Rahaman for the feedback on the draft. The authors would also like to thank NSERC, CIFAR, Google Research, Samsung, Nuance, IBM and Canada Research Chairs, Nvidia for funding, and Compute Canada for computing resources.

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

## A    PSEUDO CODE FOR GAN BASED MODEL

---

**Algorithm 2** Produce High Value States

---

**Require:** Critic $V(s)$
**Require:** $D$; transformation 'decoder' from $g$ to $s$
**Require:** Experience buffer $\mathcal{B}$ with tuples $(s_t, a_t, r_t, s_{t+1})$
**Require:** gen_state; boolean whether to generate states
**Require:** $GAN$, some generative model trained to model high-value goal states
  1: **if** gen_state **then**
  2:    $g \sim GAN$
  3:    $D : g \mapsto s$
  4:    Return $s$
  5: **else**
  6:    Return $argmax(V(s)) \; \forall s \in \mathcal{B}$
  7: **end if**

---

## B    PERFORMANCE BY VARYING LENGTH

In Figure 8 we show the performance in learning efficiency when the length of the backward traces is varied.

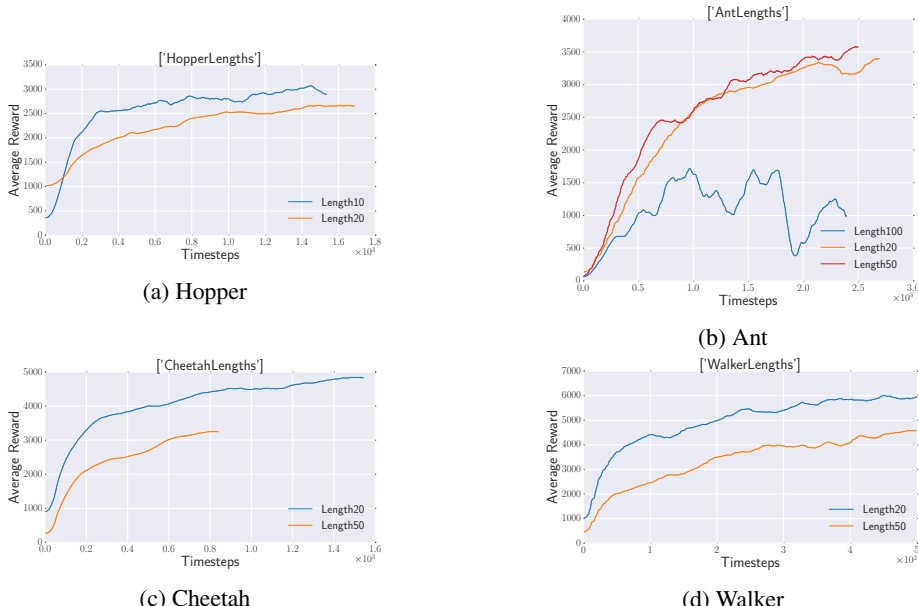

(a) Hopper

(b) Ant

(c) Cheetah

(d) Walker

Figure 8: Performance of our model (with TRPO) by varying the length of traces from backtracking model. All the time-steps are in thousands i.e (x1000)

## C    ARCHITECTURE AND IMPLEMENTATION DETAILS

The backtracking model we used for all the experiments consisted of two multi-layer perceptrons: one for the backward action predictor $Q(a_t|s_{t+1})$ and one for the backward state predictor $Q(s_t|a_t, s_{t+1})$. Both MLPs had two hidden layers of 128 units. The action predictor used hyperbolic tangent units while the inverse state predictor used ReLU units. Each network produced as output the mean and variance parameters of a Gaussian distribution. For the action predictor the output variance was fixed to 1. For the state predictor this value was learned for each dimension.

We do about a hundred training-steps of the backtracking model for every 5 training-steps of

the RL algorithm. For training the backtracking model we maintain a buffer which stores states (state, action, nextstate, reward) yielding high rewards. We sample a batch from the buffer and then normalize the states, actions before training the backtracking model on them.

During the sampling phase we feed in the normalized nextstate in the backward action predictor $Q(a_t|s_{t+1})$ to get normalized action. Then we input this normalized action in the backward state predictor $Q(s_t|a_t, s_{t+1})$ to get normalized previous state. We then un-normalize the obtained previous states and action using the corresponding mean and variance to compute the Imitation loss. This is required for stability during sampling.

# D  ADDITIONAL RESULTS

FOUR ROOM ENVIRONMENT

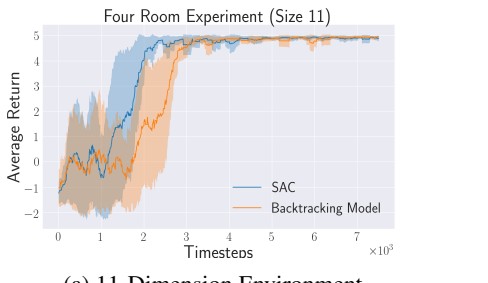
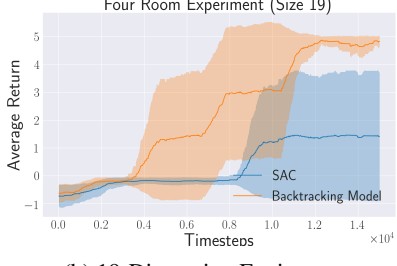

(a) 11-Dimension Environment

(b) 19-Dimension Environment

Figure 9: Training curves from the Four Room Environment for the Actor-Critic baseline (blue) and the backtracking model augmented Actor-Critic (orange). As the size of the environment increases, the benefit of the backtracking model increases for the policy. For the size-19 environment, several of the Actor-Critic baselines failed to converge, whereas the augmented recall trace model always succeeded in the number of training steps considered.

COMPARISON WITH PER

We show additional results for 13 Dimensional Environment.

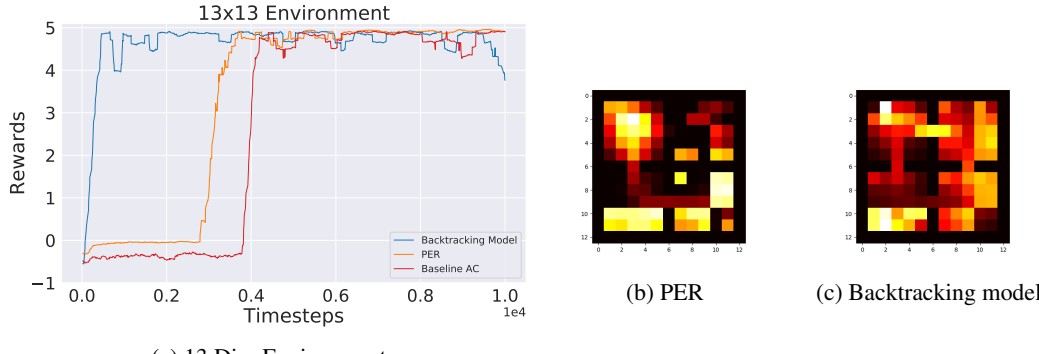

(a) 13 Dim Environment

Figure 10: Plot(a) for Reward(y) vs Timesteps(x) comparing the performance of backtracking model, PER and baseline Actor Critic. We can see that backtracking model consistently beats PER and the gap increases for increasing dimension. Heatmaps(b,c) indicating the visitation count on various positions of the grid on trained policies for the 13-Dimensional environment. We can see that the Backtracking visits a lot more states that PER does.

U-MAZE NAVIGATION

An agent has to navigate within $\epsilon-$ distance of the goal position at the end of a U-shaped maze.

For the Point Mass U-Maze navigation, we use high return trajectories for training the parameters of backtracking model (i.e those trajectories which reach sub-goals defined by Goal GAN). We sample a trajectory of length 20 from our backtracking model and the recall traces are used to improve the policy via imitation learning.

For the Ant U-Maze navigation, we train identically, only with length-50 traces.

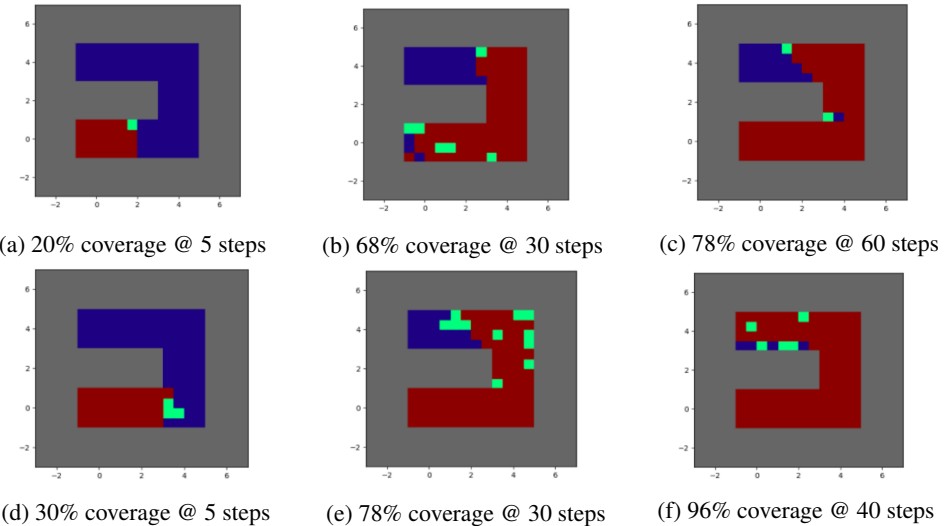

(a) 20% coverage @ 5 steps    (b) 68% coverage @ 30 steps    (c) 78% coverage @ 60 steps

(d) 30% coverage @ 5 steps    (e) 78% coverage @ 30 steps    (f) 96% coverage @ 40 steps

Figure 11: Visualization of Goal GAN baseline (top row) versus backtracking model (bottom row) for different parts of the state space for the N-Dimensional Point Mass task. Red indicates complete success; blue indicates failure. Using the backtracking model achieves greater coverage rates for the same or fewer steps of training.

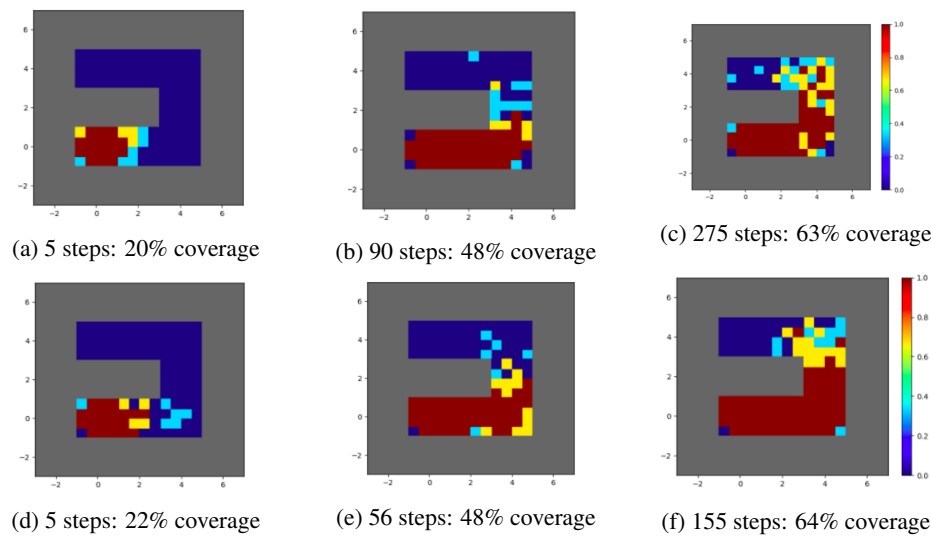

(a) 5 steps: 20% coverage    (b) 90 steps: 48% coverage    (c) 275 steps: 63% coverage

(d) 5 steps: 22% coverage    (e) 56 steps: 48% coverage    (f) 155 steps: 64% coverage

Figure 12: Visualization of Goal GAN baseline (top row) versus backtracking model (bottom row) policy performance for different parts of the state space for Ant Maze task. Red indicates complete success; blue indicates failure. Using the backtracking model achieves equal coverage rates in fewer steps of training.

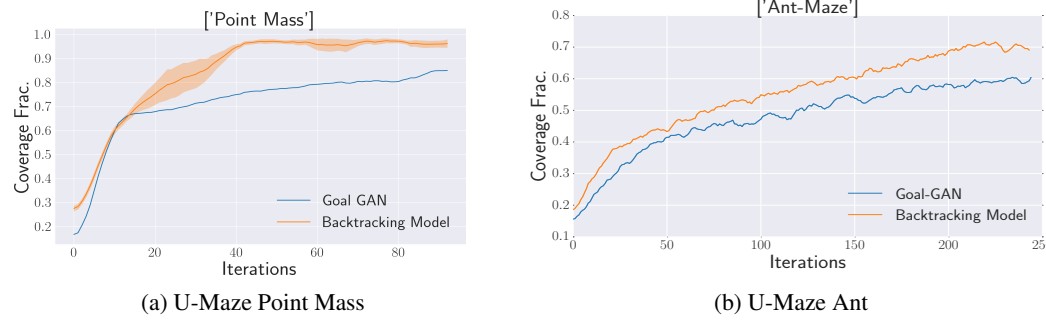

(a) U-Maze Point Mass        (b) U-Maze Ant

Figure 13: Learning curves comparing the training efficiency of our method and the baseline for both the point mass as well as ant maze task. The y-axis indicates the average return over all goal positions in the maze, and the x-axis corresponds to the number of iterations which are used for sampling goals.

# E  HYPERPARAMETERS

## E.1  ON-POLICY. TRPO

For training the backtracking model, we used an explicit buffer to store states(state, action, next-state, reward) which yields high rewards. Since, we don't have an explicit value function, we use high reward states as a proxy for high value states. We tried training the backtracking model directly from the trajectories, but it was unstable, due to changing distributions of trajectories.

| Agent | Length of trajectory Sampled from Backtracking model |
|---|---|
| Half-Cheetah | 20 |
| Walker | 20 |
| Hopper | 10 |
| Ant | 50 |

Table 1: On-Policy TRPO Hyper-params:- Length of the trajectory sampled from our backtracking model

## E.2  OFF-POLICY. SAC

For training the backtracking model, we used the high value states (under the current value function) from the buffer $\mathcal{B}$. We sample a batch of 20K high value tuples from the experience replay buffer. The length of trajectory sampled from backtracking model for each agent is shown in Table 1.

The rest of the model hyperparameters are identical to those used in Soft Actor Critic Held et al. (2017).

| Agent | Length of trajectory Sampled from Backtracking model |
|---|---|
| Half-Cheetah | 20 |
| Walker | 20 |
| Hopper | 20 |
| Ant | 50 |
| Humanoid | 50 |

Table 2: Off Policy SAC Hyperparams:- Length of the trajectory sampled from our backtracking model

## E.3  PRIORITIZED EXPERIENCE REPLAY

The Experience Replay Buffer capacity was fixed at 100k. No entropy regularization was used.

| Environment Size | Batch-size | Num. of Actor Critic steps per PER step | PER - $\alpha$ | PER $\beta$ |
|:---:|:---:|:---:|:---:|:---:|
| 11x11 | 200 | 3 | 0.8 | 0.1 |
| 13x13 | 2000 | 3 | 0.8 | 0.1 |
| 15x15 | 1000 | 3 | 0.95 | 0.1 |

Table 3: Hyperparameters for the PER Implementation.

## F  RANDOM SEARCH WITH BACKWARD MODEL

We test the performance of backtracking model when it is not learned i.e the backtracking model is a random model, in the Four Room Environment. In this experiment, we compare the scenario when the backward action predictor $Q(a_t|s_{t+1})$ is learned and when it is random. Since in the Four Room Environment, we have access to the true backward state predictor $Q(s_t|a_t, s_{t+1})$ we use that for this experiment. The comparison is shown in the figure 14. It is clear from the figure that it is helpful to learn the backward action predictor $Q(a_t|s_{t+1})$.

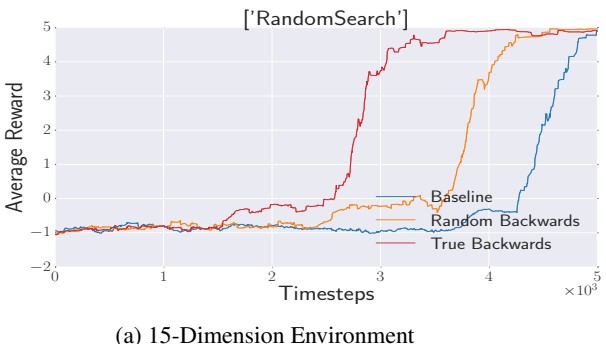

(a) 15-Dimension Environment

Figure 14: Training curves from the Four Room Environment for the Actor-Critic baseline (blue), the backtracking model augmented Actor-Critic (red), and the random search (blue).

Figure 15 shows the same comparison for Ant-v1, and Walker2d-v1 where the baseline is Soft Actor Critic (SAC). For our baseline i.e the scenario when backtracking model is not learned, we experimented with all the lengths 1,2,3,4,5, 10 and we choose the best one as our baseline. As you can see in the Figure 15, backtracking model performs best as compared to SAC baseline as well as to the scenario when backtracking model is not trained. Hence proving that backtracking model is not just doing random search.

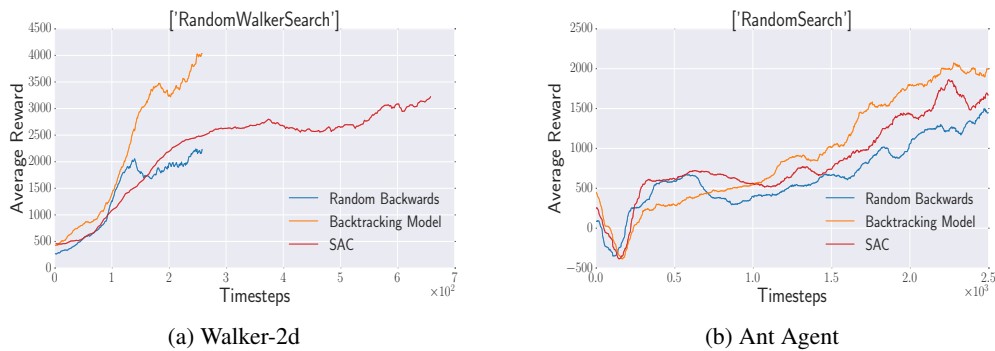

(a) Walker-2d          (b) Ant Agent

Figure 15: Training curves for the Walker2d and Ant agent. Comparing when the backtracking model is learned v/s when it's not learned.

## G    COMPARISON TO MODEL BASED RL

Dyna algorithm uses a forward model to generate simulated experience that could be included in a model-free algorithm. This method can be used to work with deep neural network policies, but performed best with models which are not neural networks (Gu et al., 2016a). Our intuition says that it might be better to generate simulated experience from backtracking model (starting from a high value state) as compared to forward model, just because we know that traces from backtracking model are good, as they lead to high value state, which is not really the case for the simulated experience from a forward model. In Fig 16, we evaluate the Forward model with On-Policy TRPO on Ant and Humanoid Mujoco tasks. We were not able to get any better results on with forward model as compared to the Baseline TRPO, which is consistent with the findings from (Gu et al., 2016a).

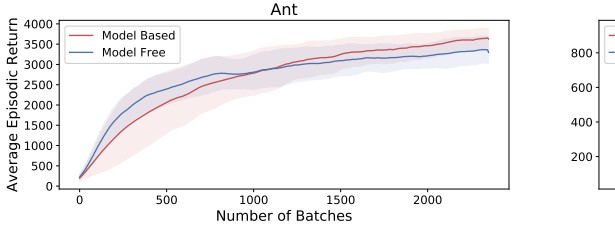 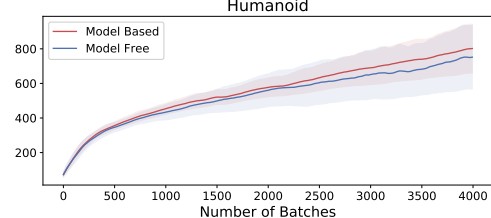

Figure 16: Forward Model compared with Baseline TRPO. We can clearly see that on Ant and Humanoid Mujoco tasks using a forward Model-based approach doesn't help.

Building the backward model is necessarily neither harder nor easier. Realistically, building any kind of model and having it be accurate for more than, say, 10 time steps is pretty hard. But if we only have 10 time steps of accurate transitions, it is probably better to take them backward model from different states as compared to from forward model from the same initial state. (as corroborated by our experiments).

Something which remains as a part of future investigation is to train the forward model and backtracking model jointly. As backtracking model is tied to high value state, the forward model could extract the goal value from the high value state. When trained jointly, this should help the forward model learn some reduced representation of the state that is necessary to evaluate the reward. Ultimately, when planning, we want the model to predict the goal accurately, which helps to optimize for this "goal-oriented" behaviour directly. This also avoids the need to model irrelevant aspects of the environment.

## H    LEARNING THE TRUE ENVIRONMENT VS LEARNING FROM RECALL TRACES

Here, we show the results of various ablations in the four-room environment which highlight the effect various hyperparameters have on performance.

In Fig 17 we train on the recall traces after a fixed number of iterations of learning in the true environment. For all of the environments, as we increase the ratio of updates in the true environment to updates using recall traces from backward model, the performance decreases significantly. This again highlights the advantages of learning from recall traces.

In Fig. 18, we see the effects of training from the recall traces multiple times for every iteration of training in the true environment. We can see that as we increase the number of iteration of learning from recall traces, we correspondingly need to choose smaller trace length. For each update in the real environment, making more number of updates helps if the trace length is smaller, and if the trace length is larger, it has a detrimental effect on the learning process as is seen in 18. Also as observed in the cases of 12x12 and 14x14 environment, it may happen that for an increased ratio of learning from recall traces and high trace length, the model achieves the maximum reward initially but after sometime the average reward plummets.

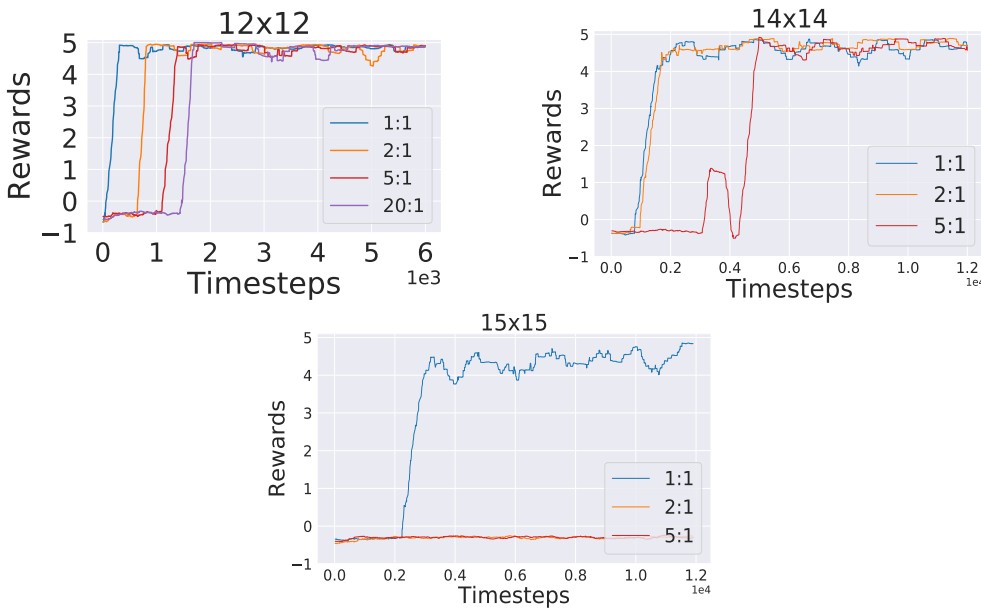

Figure 17: Legend indicates the ratio of updates in the true environment to updates using recall traces. Here we learn more in the true environment than using traces. We can see that not learning regularly from recall traces gives decreased performance.

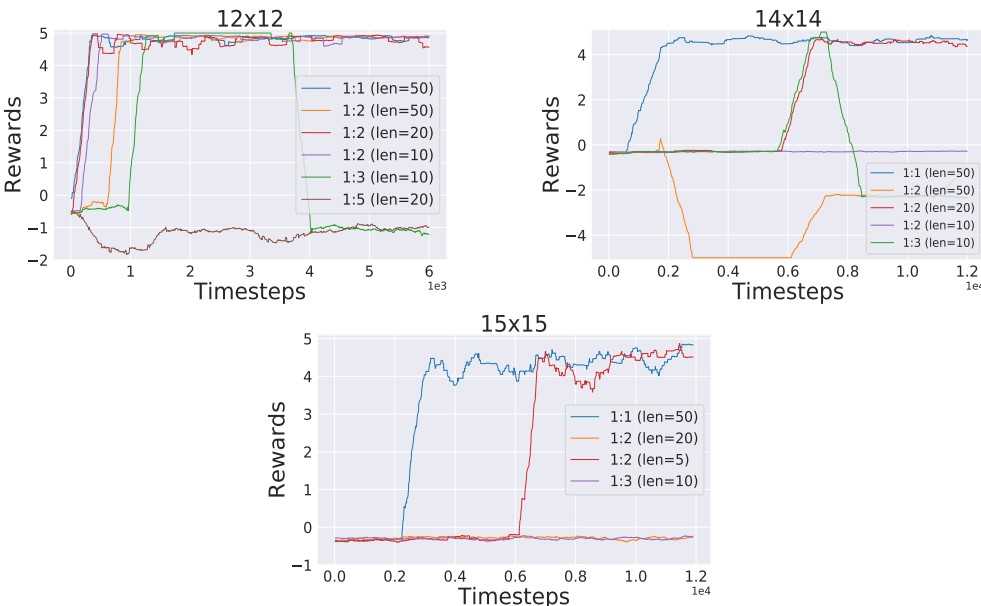

Figure 18: Legend indicates the ratio of updates in the true environment to updates using recall traces with the trajectory length of traces in parenthesis. Here we learn more using recall traces than actual environment. This tells that more updates using recall traces helps if we have a correspondingly lower trajectory length

OFF POLICY CASE FOR MUJOCO

We investigate the effect of doing more updates from the generated recall traces on some Mujoco tasks using Off-Policy SAC. As can be seen from 19, we find that using more traces helps and that for an increased number of updates we need to correspondingly shorten the trajectory length of the sampled traces.

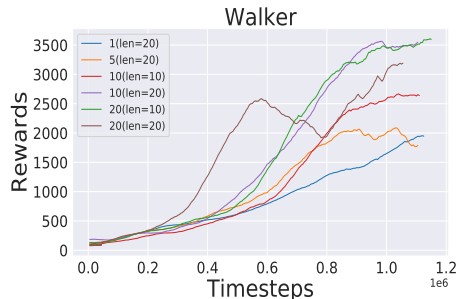 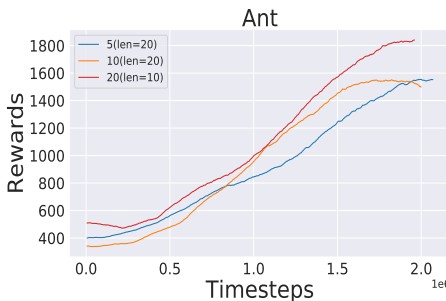

Figure 19: Legend indicates the number of updates from recall traces(per 5 updates of SAC) with the trajectory length in parentheses. In both the environments doing 10 or 20 updates is better than 5 updates. Also for 20 updates we need to choose a smaller trajectory of length 10 as compared to length 20 for 10(and 5) updates.

These experiments show that there is a balance between how much we should train in the actual environment and how much we should learn from the traces generated from the backward model. In the smaller four room-environment, 1:1 balance performed the best. In Mujoco tasks and larger four room environments, doing more updates from the backward model helps, but in the smaller four room maze, doing more updates is detrimental. So depending upon the complexity of the task, we need to decide this ratio.

