# OpenReview forum: "Recall Traces: Backtracking Models for Efficient Reinforcement Learning"
_ICLR.cc/2019/Conference_

### Official Review · AnonReviewer3 · 2018-11-02
**adding another direction to the model increases the sampling efficiency**

**Rating:** 6
**Confidence:** 3

**Review:**

The authors propose a bidirectional model for learning a policy. In particular, a backtracking model was proposed to start from a high-value state and sample back the sequence of actions and states that could lead to the current high-value state. These traces can be used later for learning a good policy. The experiments show the effectiveness of the model in terms of increase the expected rewards in different tasks. However, learning the backtracking model would add some computational efforts to the entire learning phase. I would like to see experiments to show the computational time for these components.

---

> ### Author Response · Authors · 2018-11-14
> **Thanks for your feedback!**
>
> We thank the reviewer for the positive and constructive feedback.
>
> "I would like to see experiments to show the computational time for these components."
>
> If a backtracking model model is available (like in the maze example), then there is no extra computation time, but in the case where we have to learn a bw model, learning a bw model requires more updates compared to only earning a policy (but a similar number of updates as compared to learning a forward model, i.e., dynamics model of the environment).
>
> Please let us know if anything is unclear here,  or if there is any other comparison that would be helpful in clarifying things more.

---

### Official Review · AnonReviewer2 · 2018-11-03

**Rating:** 7
**Confidence:** 3

**Review:**

This paper nicely proposes a back-tracking model that predicts the trajectories that may lead to high-value states. The proposed approach was shown to be effective in improving sample efficiency for a number of environments and tasks.

This paper looks solid to me, well-written motivation with theoretical interpretations, although I am not an expert in RL.

Comments / questions:
- how does the backtracking model correspond to a forward-model? And it doesn't seem to be contradictory to me that the two can work together.
- could the authors give a bit more explanation on why the backtracking model and the policy are trained jointly? Would it still work if to train the backtracking model offline by, say, watching demonstration?

Overall this looks like a nice paper.

---

> ### Author Response · Authors · 2018-11-14
> **Comparison to Forward Model (1/2)**
>
> The authors thank the reviewer for the positive and constructive feedback. We appreciate that the reviewer finds that our method is clearly explained.
>
> "how does the backtracking model correspond to a forward-model? And it doesn't seem to be contradictory to me that the two can work together."
>
> The reviewer raises a good point. This is indeed very useful. The Dyna algorithm uses a forward model to generate simulated experience that could be included in a model-free algorithm. This method was used to work with deep neural network policies, but performed best with models which are not neural networks (Gu et al., 2016a). Our intuition (and as we empirically show, Figure 19, Section H of Appendix) says that it might be better to generate simulated experience from a backtracking model (starting from a high value state) as compared to forward model, just because we know that traces from the backtracking model are good traces, as they lead to high value state, which is not necessarily the case for the simulated experience from a forward model.
>
> We have added Figure 16 in Appendix( Section G) where we evaluate the Forward model with On-Policy TRPO on Ant and Humanoid Mujoco tasks. We were not able to get any better results on  with forward model as compared to the Baseline TRPO, which is consistent with the findings from (Gu et al., 2016a).
>
> In essence, building the backward model is necessarily neither harder nor easier. Realistically, building any kind of model and having it be accurate for more than, say, 10 time steps is pretty hard. But if we only have 10 time steps of accurate transitions, it is probably better to take them backward model from different states as compared to from forward model from the same initial state. (as corroborated by the findings in Fig 16 of Appendix G, and Figure 19 of Appendix H).
>
> Something which remains as a part of future investigation is to train the forward model and backtracking model jointly. As the backtracking model is tied to high value states, the forward model could extract the intended goal value from the high value state. When trained jointly, this should help the forward model learn some reduced representation of the state that is necessary to evaluate the reward. Ultimately, when planning, we want the model to predict the goal accurately, which helps to optimize for this ”goal-oriented” behaviour directly. This also avoids the need to model irrelevant aspects of the environment. We also mention this in Appendix (Section G).
>
>
> [1] (Gu et al, 2016) Continuous Deep Q-Learning with Model-based Acceleration http://proceedings.mlr.press/v48/gu16.html

---

> > ### Author Response · Authors · 2018-11-14
> > **Yes, We can Train the backtracking model offline by watching demonstration. (2/2)**
> >
> > "Would it still work if to train the backtracking model offline by, say, watching demonstration?"
> >
> > Again, The reviewer raises a good point. Yes, it's possible to train the backtracking model offline by watching demonstrations. And hence, the proposed method can also be used for imitation learning. In order to show something like this, we conducted the following experiment. We trained an expert policy on Mujoco domain (Ant) using TRPO. Using the trained policy, we sample expert trajectories, and using these trajectories we learned the backtracking model in an offline mode. Now, we trained another policy from scratch, but at the same time we sample the traces from the backtracking model.  This method is about(2.5)x more sample efficient as compared to PPO, with the same asymptotic performance.  We have not done any hyperparameter search right now, and hence it should be possible to improve these results.
> >
> > We conducted additional experiments for Atari domain(Seaquest) too. For atari we trained an expert policy using a2c. And then using samples from the expert policy we learned a backtracking model.  And then we use this backtracking model for learning a new policy from scratch. This method is about(1.8)x more sample efficient as compared to A2C, with the same asymptotic performance. These results are very preliminary but it shows that it may be possible to train the backtracking model in offline mode, and use it for learning a new policy from scratch.
> >
> > Please let us know if anything is unclear here, if you’re uncertain about part of the argument, or if there is any other comparison that would be helpful in clarifying things more.

---

### Official Review · AnonReviewer1 · 2018-11-05
**Well-presented idea but evaluation seems preliminary**

**Rating:** 7
**Confidence:** 2

**Review:**

Revision:
The authors have thoroughly addressed my review and I have consequently updated my rating accordingly.

Summary:
Model-free reinforcement learning is inefficient at exploration if rewards are
sparse / low probability.
The paper proposes a variational model for online learning to backtrack
state / action traces that lead to high reward states based on best previous
samples.
The backtracking models' generated recall traces are then used to augment policy
training by imitation learning, i.e. by optimizing policy to take actions that
are taken from the current states in generated recall traces.
Overall, the methodology seems akin to an adaptive importance sampling
approach for reinforcement learning.

Evaluation:
The paper gives a clear (at least mathematically) presentation of the core idea
but it some details about modeling choices seem to be missing.
The experimental evaluation seems preliminary and it is not fully evident when
and how the proposed method will be practically relevant (and not relevant).

My knowledgable of the previous literature is not sufficient to validate the
claimed novelty of the approach.

Details:
The paper is well written and easy to follow in general.

I'm not familiar enough with reinforcment learning benchmarks to judge the
quality of the experiments compared to the literature as a whole.
Although there are quite a few experiments they seem rather preliminary.
It is not clear whether enough work was done to understand the effect of the
many different hyperparameters that the proposed method surely must have.

The authors claim to show empirically that their method can improve sample
efficiency.
This is not necessarily a strong claim as such and could be achieved on
relatively simple tests.
In the discussion the authors claim their results indicate that their approach
is able to accelearte learning on a variety of tasks, also not a strong claim.

The paper could be improved by adding a more clear explanation of the exact way
by which the method helps with exploration and how it affects finding sparse
rewards (based on e.g. Figure 1).
It seems that since only knowledge of seen trajectories can be used to generate
paths to high reward states it only works for generating new trajectories
through previously visited states.

Questions that could be clarified:
- It is not entirely obvious to me what parametric models are used for the
backtracking distributions.
- Does this method not also potentially hinder exploration by making the agent
learn to go after the same high rewards / Does the direction of the variational
problem guarantee coverage of the support of the R > L distribution by samples?
- What would be the effect of a hyperparameter that balances learning the recall
traces and learning the true environment?
- Are there also reinforcement learning tasks where the proposed methods'
improvement is marginal and the extra modeling effort is not justified (e.g.
due to increase complexity).

Page 1: iwth (Typo)
Page 2: r(s_t) -> r(s_t, a_t)
Page 6: Prioritize d (Typo)

---

> ### Author Response · Authors · 2018-11-14
> **Thanks for your feedback!  (1/3)**
>
> Thanks for the very thorough feedback. We have conducted additional experiments to address the concerns raised about the evaluation, and we clarify specific points below. We believe that these additions address all of your concerns about the work, though we would appreciate any additional comments or feedback that you might have.
>
> "I'm not familiar enough with reinforcement learning benchmarks to judge the quality of the experiments compared to the literature as a whole."
>
> The goal of our experimental evaluation is to demonstrate the effectiveness of the proposed algorithm.  We demonstrate that the effectiveness by comparing the proposed algorithm in case when the true backtracking env. was avaliable, as well as when we learned the backtracking model too. We compare our methods to the state-of-the-art SAC algorithm on MuJoCo tasks in OpenAI gym (Brockman et al., 2016) and in rllab (Duan et al., 2016). We use SAC as a baseline as it notably outperforms other existing methods like DDPG, Soft-Q Learning and TD3. The results show that our method outperform on par with SAC in simple domains like swimmer, walker etc. They also provide evidence that the proposed method outperform SAC in challenging high dimensional domains like humanoid and Ant (Figure 7, Main Paper).
>
> "It is not entirely obvious to me what parametric models are used for the backtracking distributions."
>
>
> The backtracking model we used for all the experiments consisted of two multi-layer perceptrons: one for the backward action predictor Q(a_t | s_t+1) and one for the backward state predictor Q(s_t | a_t, s_t+1). Both MLPs had two hidden layers of 128 units. The action predictor used hyperbolic tangent units while the inverse state predictor used ReLU units. Each network produced as output the mean and variance parameters of a Gaussian distribution. For the action predictor the output variance was fixed to 1. For the state predictor this value was learned for each dimension. We have also mentioned this in the appendix.

---

> > ### Author Response · Authors · 2018-11-14
> > **Effect of hyperparameter(s) (2/3)**
> >
> > >>  What would be the effect of a hyperparameter that balances learning the recall traces and learning the true environment? >> whether enough work was done to understand the effect of the many different hyperparameters that the proposed method surely must have.
> >
> > In order to address reviewer’s question, we did more experiments on four room maze as well as on mujoco domain.
> > We have 3 parameters associated.
> > 1) How many traces to sample from backtracking model.
> > 2) How many steps each trace should be sampled for i.e is the length of the trajectory sampled.
> > 3) And as the reviewer pointed out, the effect of a hyperparameter that balances learning the recall traces and learning the true environment.
> >
> > Q1) How many traces to sample from backtracking model.
> >
> > For most of our experiments, we sample only single a trace from the backtracking model. But we observe that sampling more traces actually helps for more complex environments. This is also again in contrast as compared to the forward model.  .
> >
> > Q2)  How many steps each trace should be sampled for ?
> > In practice, if the agent is limited to one or a few initial states, a concern related to the length of generated backward traces is that longer traces become increasingly likely to deviate significantly from the traces that the agent can generate from its initial state. Therefore, in our experiments, we sample fairly short traces.  Figure 8 (Appendix, Section B) shows the Performance of our model (with TRPO) by varying the length of traces from backtracking model. All the time-steps are in thousands i.e (x1000). As evident by the figure, sampling very long traces seems to hinder the performance on all the domains.
> >
> > Q3) Effect of a hyperparameter that balances learning the recall traces and learning the true environment
> >
> > We have added a Section H in the Appendix containing ablations for the four-room environment and some Mujoco tasks which tells about the effect this hyperparameter has on effective performance.
> >
> > In Figure 17(Appendix, Section H) we noticed that as we increase the ratio of updates in the true environment to updates using recall traces from the backward model, the performance decreases. This highlights again the advantages of learning from the recall traces. In the second experiment, we see the effect of training from the recall traces multiple times for every iteration of training in the true environment. Figure 18(Appendix, Section H) shows that as we increase the number of iterations of learning from recall traces, we correspondingly need to choose a smaller trace length. For each update in the real environment, making more number of updates from recall traces helps if the trace length is smaller, and if the trace length is larger, it has a detrimental effect on the learning process.
> >
> > In Figure 19(Appendix, Section H) we again find that for Mujoco tasks doing more updates using the recall traces is beneficial. Also for more updates we need to choose smaller trajectory length.
> >
> > In essence, there is a balance between how much we should train in the actual environment and how much we should learn from the traces generated from the backward model. In the smaller four room-environment, 1:1 balance performed the best. In Mujoco tasks and larger four room environments, doing more updates from the backward model helps, but in the smaller four room maze, doing more updates is detrimental. So depending upon the complexity of the task, we need to decide this ratio.

---

> > > ### Author Response · Authors · 2018-11-14
> > > **Exploration and complexity of backtracking model  (3/3)**
> > >
> > > >> Are there also reinforcement learning tasks where the proposed methods' improvement is marginal and the extra modeling effort is not justified (e.g. due to increase complexity).
> > >
> > > We think that having a backtracking model could always improve the performance. As We evaluate it on a large number of very different domains (when the backtracking model is given as well as when we are learning the backtracking model as in off policy case and on-policy case)  and find that in all cases it improves performance. But we also think, that for some environments the backtracking model can be very hard to learn. For other problems, learning a model of the environment is difficult in either direction so those problems would be hard as well. The first issue would be severe if the forward dynamics are strongly many-to-one, for example. The second case applies to any complex environment and especially partially observed ones. Our method shines most when the dynamics are relatively simple but the problems are still hard due to sparse rewards.
> > >
> > > On the other hand, the backtracking model could also be used in practical settings like robotics, that involve repeatedly attempting to solve a particular task, and hence resetting the environment between different attempts. Here, we can use a model that learns both a forward policy and a backtracking model, and resetting of the environment can be approximated using the backtracking model. By learning this backtracking model, we can also determine when the policy is about to enter a non-reversible state, and hence can be useful for safety. It remains future work, to investigate this.
> > >
> > > >> Does this method not also potentially hinder exploration by making the agent learn to go after the same high rewards / Does the direction of the variational problem guarantee coverage of the support of the R > L distribution by samples?
> > >
> > > This is a tricky subject and it is hard to come up with principles that will improve exploration in general and to be sure that something doesn't hinder exploration for some problems. In our setup, the exploration comes mostly from the goal generation methods. The backwards model helps more to speed up the propagation of high value to nearby states (indirectly), such that fewer environment interactions are needed but that could perhaps lead to fewer trips to locations with incorrectly assumed low value. On the other hand, the method might cause the exploration of different (better) paths to the same high value states as well, which should be a good thing. In general, since we are seeking high value (i.e. high expected return), so it shouldn't hinder exploration much. But instead if we seek “high reward” states, then it would hinder performance, (as our experiments show).
> > >
> > >
> > > Closing:
> > > Thank you for your time. We hope you find that our revision addresses your concerns.
> > > Please let us know if anything is unclear here, if you’re uncertain about part of the argument, or if there is any other comparison that would be helpful in clarifying things more.

---

> > > > ### Author Response · Authors · 2018-11-22
> > > > **Thanks for your time! :)**
> > > >
> > > > We would appreciate it if the reviewer could take another look at our changes and additional results, and let us know if the reviewer would like to request additional changes that would alleviate reviewers concerns. We hope that our updates to the manuscript address the reviewer's concerns about clarity, and we hope that the discussion above addresses the reviewer's concerns about empirical significance. We once again thank the reviewer for the thorough feedback of our work.

---

> ### Author Response · Authors · 2018-11-25
> **Request for feedback ?**
>
> Thank you again for the thoughtful review. We would like to know if our rebuttal (see below, "Thanks for your feedback! (n/3) ") adequately addressed your concerns. We would also appreciate any additional feedback on the revised paper. Are there any other aspects of the paper that you think could be improved?

---

### Author Response · Authors · 2018-11-19
**Paper Updated to address reviewer feedback.**

We have updated the paper with the following changes to address reviewer comments:

- Added comparison to forward model (Reviewer 2)
- Conducted preliminary experiments to show that the backtracking model can be trained just by using the demonstrations. (Reviewer  2)
- Effect of the 3 hyperparameter(s) associated with the proposed model.

Thank you for your time! The authors appreciate the time reviewers have taken for providing feedback. which resulted in improving the presentation of our paper. Hence,  we would appreciate it if the reviewers could take a look at our changes and additional results, and let us know if they would like to either revise their rating of the paper, or request additional changes that would alleviate their concerns.

---

### Author Response · Authors · 2018-11-27
**Final Response**

We thank the reviewers for the detailed feedback on our paper. We are glad that the reviewers found our paper to be  "solid  contribution with well-written motivation with theoretical interpretations" (reviewer 2) and "well written in general" (reviewer 1).

We made the following changes to the manuscript to address the reviewers comments.

- We conducted more ablation experiments for the 3 hyper-parameters associated with our model, as asked by the Reviewer 1.

- Training backtracking model by using demonstrations (Reviewer 2)  and then using the backtracking model for training another policy from scratch. We did experiments on Ant env from mujoco and Seaquest from atari, where we first train a backtracking model from the expert demonstrations, and then use that for training policy. We achieve 2.5x and about 2x sample efficiency in our very preliminary experiments.

- Comparison with the forward model (Section G and H) as pointed by Rev 2. Rev 2 mentioned an interesting point of training forward and backward model.  Our conclusion is building the backward model is necessarily neither harder nor easier. Realistically, building any kind of model and having it be accurate for more than, say, 10 time steps is pretty hard. But if we only have 10 time steps of accurate transitions, it is probably better to take them backward.

We feel that by conducting extra experiments have improved the quality of the paper a lot, and we are grateful to reviewers for  very useful feedback.

---

### Meta-Review · Area_Chair1 · 2018-12-13
**Novel take on model-based improvement on model-free RL**

**Confidence:** 4
**Recommendation:** Accept (Poster)

**Metareview:**

The paper presents "recall traces", a model based approach designed to improve reinforcement learning in sparse reward settings. The approach learns a generative model of trajectories leading to high-reward states, and is subsequently used to augment the real experience collected by the agent. This novel take on combining model-based and model-free learning is conceptually well motivated and is empirically shown to improve sample efficiency on several benchmark tasks.

The reviewers noted the following potential weaknesses in their initial reviews: the paper could provide a clearer motivation of why the proposed approach is expected to lead to performance improvements, and how it relates to learning (and uses of) a forward model. Details of the method, e.g., model parameterization is unclear, and the effect of hyperparameter choices is not fully evaluated.

The authors provided detailed replies to all reviewer suggestions, and ran extensive new experiments, including experiments to address questions about hyperparameter settings, and an entirely new use of the proposed model in a learning from demonstration setting. The authors also clarified the paper as requested by the reviewers. The reviewers have not responded to the rebuttal, but in the AC's assessment their concerns have been adequately addressed. The reviewers have updated their scores in response to the rebuttal, and the consensus is to accept the paper.

The AC notes that the authors seem unaware of related work by Oh et al. "Self Imitation Learning" which was published at ICML 2018. The paper is based on a similar conceptual motivation but imitates high-value traces directly, instead of using a generative model. The authors should include a discussion of how their paper relates to this earlier work in their camera ready version.